# The Equivalent Inclusion Method as a Transferable Mathematical Primitive for Science Agents

## Abstract

We formalize the *Equivalent Inclusion Method* (EIM) as an operator–theoretic primitive that an autonomous science agent can apply uniformly across disciplines. Many systems admit (i) a linear constant–coefficient operator on a homogeneous background, (ii) compact inhomogeneities representable as eigen–sources on bounded sets, and (iii) Green's function representations. Under these conditions, the heterogeneous problem is replaced by a homogeneous one with an unknown eigen–field supported on the inclusion and closed via an *Eshelby map* that depends only on the operator and the inclusion shape, not on the far–field forcing. We derive this machinery for reaction–diffusion–advection (RDA) dynamics, obtain a generalized Eshelby map and screened–Laplace limits, provide the two–inclusion interaction law, and develop analytical effective–medium closures (dilute/Maxwell–Garnett, Mori–Tanaka, and self–consistent) for the composite growth rate. Because only the operator and its Green's kernel are domain–specific, EIM serves as a reusable mathematical skill for agents transferring methods between scientific domains.

## 1 Introduction

**Motivation.** Autonomous scientific agents benefit from portable, operator–centric skills that transfer across domains with minimal adaptation. The *Equivalent Inclusion Method* (EIM) is such a skill. Originating in elasticity with Eshelby's celebrated discovery that an ellipsoidal inclusion subjected to uniform eigenstrain produces a uniform interior strain [**1957a**, **1959a**], EIM replaces material heterogeneity by fictitious eigenfields on bounded supports and recovers the physical fields through the background Green's function [**Mura1987**]. The same closure principle extends well beyond linear elasticity: to steady heat conduction and transport [**Hiroshi1986**, **Hatta1986**, **Yin2005**, **Yin2008b**], to electro/magnetostatics and coupled multi-physics [**Lu2011**], and to computational homogenization and numerical variants [**Nakasone2000**, **Brisard2014**, **Otero2015**, **Sakata2008**, **Sakata2010**, **Kushch2016**].

**From mechanics to operators.** Viewed through an operator lens, EIM relies on three ingredients: a constant-coefficient linear operator on a homogeneous background, compact inhomogeneities representable as eigen–sources, and a Green's representation. These enable a Lippmann–Schwinger formulation and an interior *Eshelby map* that depends only on the operator and inclusion geometry, not on the far field. For ellipsoids, the interior field is uniform (Eshelby property) [**1957a**, **1959a**]; for non-ellipsoids, intrinsic non-uniformity or singular behavior can arise, with many constructive generalizations available for polygons and polyhedra [**Rodin1996**, **Nozaki1997**, **Nozaki2000**, **Trotta2016**, **Trotta2017**, **Liu2013**, **Wu2021a**, **Wu2021b**].

**Numerical realizations.** Boundary integral and inclusion-based discretizations assemble these closures efficiently by reusing the same kernels across problems [**Lachat1976**, **Gaul2003**, **2008a**,

---

**Wu2021**, **Zhou2014**, **Dong2002**]. This paper adopts the same reuse principle for reaction–diffusion–advection (RDA) dynamics: once the screened Green's kernel is known, inclusion thresholds, interior amplification, and interaction laws follow from geometry-only Eshelby maps. The screened (Yukawa) structure connects naturally to classical potential theory and half-space kernels [**Thomson1848**, **Boussinesq1885**, **Rosati2014**].

**Contributions.** We make five contributions: (1) an operator–theoretic formalization of EIM with a Lippmann–Schwinger backbone; (2) closed-form space–time and resolvent Green's functions for RDA with advection; (3) generalized Eshelby maps at monopole/dipole order and screened–Laplace limits; (4) a two–inclusion interaction determinant and a multi–inclusion kernel eigenproblem; and (5) analytical effective–medium closures (dilute/Maxwell–Garnett, Mori–Tanaka, self–consistent) for composite growth in RDA. Related work in elasticity, potential problems, and inclusion-based BEM is summarized in sec:related [**1957a**, **1959a**, **Mura1987**, **Hiroshi1986**, **Nakasone2000**, **Brisard2014**, **Zhou2014**, **Wu2021**].

**Roadmap.** sec:prelim establishes notation and the operator setting. sec:master presents the master EIM equations and Eshelby maps. sec:rda derives RDA kernels and screened limits. sec:single analyzes single-inclusion thresholds and amplification. sec:multi develops two- and multi-inclusion interactions. sec:effective presents analytical effective-medium closures, and sec:workflow distills an agent-facing workflow. Cross-domain transfer is sketched in sec:map, with related work, limitations, and conclusions in sec:related,sec:disc,sec:concl.

## 2 Preliminaries and notation

Let the field be $u : \mathbb{R}^n \to \mathbb{C}^m$ governed by a linear constant–coefficient operator $\mathcal{L}$ acting on a homogeneous background. A compact inclusion $V \subset \mathbb{R}^n$ modifies parameters from background $C_0$ to $C_1$. Denote the indicator by $\chi_V$, the inclusion centroid by $x_c$, and the free-space Green's tensor by $G$ satisfying $\mathcal{L} G = \delta$ with the chosen radiation/causality condition. Convolution is written $(G * s)(x) = \int G(x - y) \, s(y) \, \mathrm{d}y$.

Advection $v$ in RDA is removed by a standard gauge/shift, producing a screened parameter $\kappa$ recorded in eq:screened.

## 3 Operator–theoretic EIM (master formulation)

We pose the heterogeneous problem as

$$\mathcal{L}u = f + \chi_V s^*, \qquad s^* = \mathsf{A}\, e \ or \ s^* = \mathsf{N}\, u, \tag{1}$$

which admits the Lippmann–Schwinger representation

$$u = u^\infty + \int_V G\, s^*, \tag{2}$$

under standard existence/uniqueness assumptions. An interior *Eshelby map* $\mathsf{S}$ closes the inclusion:

$$e_{\mathrm{in}} = e^\infty + \mathsf{S}\, e^*, \qquad e^* = \mathsf{A}\, e_{\mathrm{in}}, \tag{3}$$

so that $e_{\mathrm{in}} = (\mathsf{I} - \mathsf{SA})^{-1} e^\infty$. At monopole and dipole order, $\mathsf{S}_0 = \int_V G(x - y) \, \mathrm{d}y$, $\mathsf{S}_1 = \int_V (\nabla_x G) \otimes (y - x_c) \, \mathrm{d}y$. For ellipsoids, $e_{\mathrm{in}}$ is uniform [**1957a**, **1959a**, **Mura1987**]; loss of invertibility, $\det(\mathsf{I} - \mathsf{SA}) = 0$, signals an unforced eigenmode (threshold).

## 4 RDA epidemics: Green's functions and screened limits

**Model.** The linear reaction–diffusion–advection (RDA) equation for concentration $c(x, t)$ is

$$\partial_t c = D\, \nabla^2 c - v \cdot \nabla c + a\, c + f(x, t), \qquad D > 0. \tag{4}$$

**Space–time kernel.** With the Galilean shift $y = x - vt$, the causal kernel is the advected, growing heat kernel

$$G(x, t) = H(t)\, (4\pi Dt)^{-n/2} \exp\!\big(at - \|x - vt\|^2/(4Dt)\big), \tag{5}$$

which satisfies $(\partial_t - D\nabla^2 + v \cdot \nabla - a)G = \delta(t)\, \delta(x)$.

**Resolvent kernels.** After Laplace transform in time (parameter $s$), one obtains a modified Helmholtz (Yukawa) problem with

$$\mu^2 = (s - a) + \frac{\|v\|^2}{4D}, \qquad \kappa = \mu/\sqrt{D}. \tag{6}$$

The resolvent $\tilde{G}(x;s)$ is 1D: $\quad \tilde{G}(x;s) = \frac{1}{2\sqrt{D}\,\mu} \exp\left(v \cdot x2D - \kappa\,|x|\right),$

$2D: \quad \tilde{G}(x;s) = \frac{1}{2\pi D} \exp\left(v \cdot x2D\right) K_0(\kappa r), \ r = \|x\|,$

$3D: \quad \tilde{G}(x;s) = \frac{1}{4\pi Dr} \exp\left(v \cdot x2D - \kappa r\right)$. The steady/resolvent limit $s = 0$ is a screened–Laplace (Yukawa) kernel with rate $\kappa$; anisotropic $D$ is handled by an affine change of metric [**Rosati2014**].

# 5 Single-inclusion analysis: thresholds and amplification

**Contrast and closure.** Let $a(x) = a_0 + \Delta a\,\chi_V(x)$. In the unforced (eigenmode) case, the monopole closure yields the secular law

$$1 - \Delta a\,S_0(\kappa; V) = 0, \qquad S_0(\kappa; V) = \int_V G_\kappa(x - y)\,\mathrm{d}y, \tag{7}$$

where $G_\kappa$ is the steady screened kernel. For ellipsoids, $S_0$ is interior-constant (uniform Eshelby map), so the threshold depends only on $\kappa$ and geometry [**1957a**, **1959a**, **Mura1987**, **Jin2011**].

**Critical sizes.** In 1D, a classical estimate near threshold gives $L_{\mathrm{crit}} \approx \pi\sqrt{D/a_1}$ when the exterior is subcritical ($a_0 < 0$). In 2D and 3D, disks/spheres yield Bessel/Yukawa integrals relating $R$ and $\kappa$; small $\kappa R$ increases the needed $R$, while large $\kappa R$ approaches unscreened depolarization factors.

**Forced problems.** With a background drive $u^\infty$, the interior response obeys

$$u_{\mathrm{in}} = (1 - \Delta a\,S_0)^{-1}\,u^\infty, \tag{8}$$

explicitly quantifying hotspot magnification.

# 6 Two- and multi-inclusion interactions

For disjoint $V_1, V_2$ (centers $x_1, x_2$, volumes $|V_i|$), monopole order gives

$$1 - \Delta a_1 S_{0,1} - \Delta a_1 |V_1| S_{12} - \Delta a_2 |V_2| S_{21} 1 - \Delta a_2 S_{0,2} u_1 u_2 = 0, \tag{9}$$

with $S_{ij} \approx G_\kappa(|x_i - x_j|)$. A nontrivial mode exists iff

$$(1 - \Delta a_1 S_{0,1})(1 - \Delta a_2 S_{0,2}) = \Delta a_1 \Delta a_2\,|V_1||V_2|\,S_{12}^2. \tag{10}$$

In 2D, $S_{12} \propto K_0(\kappa d)$; in 3D, $S_{12} \propto e^{-\kappa d}/d$, so decreasing separation $d$ lowers each patch's critical size (cooperation). Multi-inclusion persistence corresponds to the smallest eigenvalue of the symmetric kernel matrix crossing zero.

# 7 Analytical effective-medium closures for RDA

**Setup (drift-removed resolvent).** In the drift-removed frame $\psi = \exp(-v \cdot x/(2D))\,\Phi$,

$$(-D\nabla^2 + \mu^2)\psi = -\sum_j \Delta a_j\,\chi_{\Omega_j}\,\psi, \qquad \mu^2 = (s - a_0) + \frac{\|v\|^2}{4D}, \tag{11}$$

so the 2D free-space resolvent is $\widehat{G}(r;s) = 12\pi D K_0(\kappa r)$ with $\kappa = \mu/\sqrt{D}$ (cf. eq:resolvent2d,eq:screened). For a single inclusion family (volume fraction $f$, radius $R$, contrast $\Delta a$), the generalized Eshelby tensor reduces to a scalar for a disk,

$$S_0(s; R) = \frac{1}{\pi D}\left[\frac{1}{(\kappa R)^2} - \frac{K_1(\kappa R)}{\kappa R}\right], \qquad z = \kappa R. \tag{12}$$

**(1) Dilute (Maxwell–Garnett–type) effective growth.** In the dilute limit (non-interacting inclusions),

$$\Delta a_{\text{eff}}^{\text{dil}}(s) = f \, \frac{\Delta a}{1 - \Delta a \, S_0(s; R)} \; . \tag{13}$$

Thus $a_{\text{eff}}(s) = a_0 + \Delta a_{\text{eff}}^{\text{dil}}(s)$ and $\mu_{\text{eff}}^2(s) = s - a_{\text{eff}}(s) + \|v\|^2/(4D)$. For $z \ll 1$,

$$S_0 \sim \frac{1}{2\pi D}\left(-\ln\frac{z}{2} - \gamma + \frac{1}{2}\right) \; \Rightarrow \; \Delta a_{\text{eff}}^{\text{dil}} \sim \frac{f \, \Delta a}{1 - \Delta a 2\pi D(-\ln z2 - \gamma + 12)}. \tag{14}$$

**(2) Mori–Tanaka (interaction-renormalized).** A matrix-averaged concentration factor yields a closed form at finite $f$:

$$\Delta a_{\text{eff}}^{\text{MT}}(s) = \frac{f \, \Delta a}{1 - (1 - f)\,\Delta a \, S_0(s; R)} \; . \tag{15}$$

It reduces to the dilute law as $f \to 0$ and softens the divergence as $f \uparrow 1$.

**(3) Self-consistent (multiple-scattering averaged).** Replacing the matrix by the unknown effective medium, let $a_i = a_0 + \Delta a$, $\kappa_{\text{eff}}(s) = \sqrt{(s - a_{\text{eff}}) + \|v\|^2/(4D)}/\sqrt{D}$, and evaluate $S_0$ at $\kappa_{\text{eff}}$. Then

$$a_{\text{eff}}(s) = a_0 + \frac{f\,(a_i - a_{\text{eff}})}{1 - (a_i - a_{\text{eff}})\,S_0(s; R)\big|_{\kappa \to \kappa_{\text{eff}}(s)}} \; . \tag{16}$$

This scalar nonlinear equation is easily solved by fixed point or Newton.

**(4) Composite growth rate.** Neutrality occurs when $\mu_{\text{eff}}^2(\lambda_{\text{eff}}) = 0$, i.e.

$$\lambda_{\text{eff}} = a_{\text{eff}}(\lambda_{\text{eff}}) - \frac{\|v\|^2}{4D} \; . \tag{17}$$

In dilute/MT, use the explicit $a_{\text{eff}}(s)$; in SC, substitute the self-consistent relation into the same condition.

**(5) Beyond circular inclusions.** For ellipses (2D) or oriented families, replace $S_0$ by the appropriate contraction of the generalized Eshelby tensor $S$ with the uniform interior mode; orientation distributions enter via averaging. If inclusions also perturb $D$ or $v$, first- and second-order blocks of $S$ generate anisotropic $D_{\text{eff}}$ and a corrected drift; for pure "growth hotspots," $D$ and $v$ are unchanged at leading order (they enter through $\kappa$).

# 8 Agent-facing workflow (deterministic template)

1. Normalize operator (gauge out $v$; nondimensionalize).
2. Select kernel $G$ (Laplace/screened/Helmholtz/etc.).
3. Assemble Eshelby maps $S_0$, $\mathsf{S}_1$ for the geometry.
4. Close and solve: threshold, interactions, interior amplification.
5. Perform analytical checks: small/large $\kappa R$ asymptotics; sensitivity of $a_{\text{eff}}$ and $\lambda_{\text{eff}}$ to $f$, $R$, and $\Delta a$.
6. Emit parameters and a figure checklist (threshold curves, interaction laws, effective-medium predictions).

# 9 Cross-domain operator map

# 10 Related work

EIM in elasticity descends from Eshelby and systematic micromechanics [**1957a**, **1959a**, **Mura1987**]; polygonal and polyhedral generalizations refine interior closures [**Rodin1996**, **Nozaki1997**, **Nozaki2000**, **Trotta2016**, **Trotta2017**, **Liu2013**, **Wu2021a**, **Wu2021b**]. Variational and numerical

Table 1: Domains admitting EIM closures (examples).

| Domain | Representative PDE | Eigen-quantity | Why EIM applies |
|--------|--------------------|----------------|-----------------|
| Electrostatics | $\nabla \cdot (\varepsilon \nabla \phi) = -\rho$ | Polarization/$\rho^*$ | Permittivity contrast $\Rightarrow$ equiv. polarization |
| Magnetostatics | $\nabla \cdot (\mu \nabla \psi) = 0$ | Magnetization | Permeability contrast $\Rightarrow$ magnetization |
| Thermal (steady) | $\nabla \cdot (k \nabla T) = -q$ | Eigen-flux/$q^*$ | Conductivity contrast $\Rightarrow$ depolarization tensors |
| Elasticity (static) | Navier–Lamé | Eigenstrain $\varepsilon^*$ | Eshelby inclusion; uniform interior for ellipsoids [19 |
| Stokes flow | Steady Stokes | Eigen strain-rate/body force | Viscosity/density contrasts as eigen-sources [**Yin201** |
| Hydrogeology | Laplace/Poisson | Pumping/injection | Heterogeneity as equivalent source/sink |

EIM support homogenization [**Brisard2014**, **Otero2015**, **Sakata2008**, **Sakata2010**, **Kushch2016**]. Inclusion-based and boundary element methods (BEM) realize these closures efficiently and robustly [**Lachat1976**, **Gaul2003**, **2008a**, **Dong2002**, **Zhou2014**, **Wu2021**]. Analogues in steady transport and screened operators connect to classical potential theory and Yukawa responses [**Hiroshi1986**, **Hatta1986**, **Rosati2014**].

# 11 Discussion and limitations

EIM provides robust *sign* predictions near threshold (growth vs. decay). Quantitative magnitudes depend on domain boundaries, anisotropy, and parameter priors. Drift and anisotropic diffusion enter through the screened metric; highly asymmetric multi-inclusion arrangements can weakly hinder. Nonlinear saturation (e.g., SIR-type kinetics) requires extensions beyond linear closures.

# 12 Conclusion

EIM functions as a compact, reusable operator primitive. With RDA kernels in hand, inclusion thresholds, amplification, and interaction laws follow from geometry-only Eshelby maps, enabling agent transfer across domains.

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
