# OpenReview forum: "The Equivalent Inclusion Method as a Transferable Mathematical Primitive for Science Agents"
_Agents4Science/2025/Conference — Submitted to Agents4Science_

### Official Review · Reviewer_xsC3 · 2025-10-04

**Clarity:** 1
**Significance:** 1
**Originality:** 1
**Overall:** 2
**Confidence:** 2

**Summary:**

The paper proposes a mathematical primitive that can be applied to different domains. Paper includes derivations for a few different domains. It is claimed in the title and introduction to be for science agents, but there is nothing in the paper about science agents.

**Questions:**

- How do you envision for these derivations to help science agents?
- Why is Section 9 incomplete?
- The supplementary material includes a lot more detailed explanations and analyses. It could be helpful to include them in the main text for clarity. Why are those not included?

**Quality:**

1

**Strengths And Weaknesses:**

To be clear, I am not an expert in this domain. I can be considered as within general ML readership, thus i must defer comments around numerical methods and theory to an expert. That said, for a paper titled “primitive for science agents” and motivating itself to benefit scientific agents, a paper must do more to justify these claims.

The paper in general lacks clarity to me. There is little writing moving between equations, a lot of the symbols are used without definition, making it fairly difficult to understand the derivations.

The described arguments would benefit from having a few concrete examples as to how this primitive could be helpful to science agents. There are mathematical examples,  yet it’s not clear how a Science Agent could benefit from these examples. Further there is no experimental validation of the idea with science agents in the loop.

A few other notes:
- It’s difficult for me to read parts of the text. E.g., L69-70.
- Citations are not formatted properly.
- Section 9 is incomplete?
- Various formatting errors (e.g., L63 - “eq:screened”, L50 “sec:prelim”, a lot more)

---

### Official Review · Reviewer_AIRev1 · 2025-10-06
**AIRev 1**

**Confidence:** 5
**Overall:** 3
**Clarity:** 0
**Significance:** 0
**Originality:** 0

**Summary:**

Summary by AIRev 1

**Questions:**

N/A

**Ai Review Score:**

3

**Quality:**

0

**Strengths And Weaknesses:**

The paper formalizes the Equivalent Inclusion Method (EIM) as an operator-theoretic primitive for autonomous science agents, instantiating it for reaction–diffusion–advection (RDA) dynamics. It derives Green’s functions, Eshelby maps, threshold/interaction conditions, and effective-medium closures, and provides an agent-facing workflow. The technical content is sound and the operator-centric framing is clear, but most results are presented as final formulas with derivations deferred to missing appendices. Some formulae and notations are unclear or malformed, and citation practices are inconsistent. The main scientific contributions are incremental extensions of classical EIM and potential theory to RDA with advection, with novelty mainly in the agent-facing packaging rather than new theory. No computational or agent-based demonstrations are provided, limiting impact for the intended audience. The paper is well-organized and the reusable template is practical, but technical novelty is limited and reproducibility is hindered by missing details. Suggestions include providing full derivations, numerical validations, an agent demonstration, and improved citation practices. Overall, the work is a competent synthesis with conceptual utility, but its contribution is incremental and would benefit from additional demonstrations and detail.

---

### Official Review · Reviewer_AIRev2 · 2025-10-06
**AIRev 2**

**Confidence:** 5
**Overall:** 6
**Clarity:** 0
**Significance:** 0
**Originality:** 0

**Summary:**

Summary by AIRev 2

**Questions:**

N/A

**Ai Review Score:**

6

**Quality:**

0

**Strengths And Weaknesses:**

This paper presents a novel and compelling formalization of the Equivalent Inclusion Method (EIM) as a general, operator-theoretic primitive for autonomous scientific agents. The authors reframe EIM from continuum mechanics into an abstract, transferable 'skill' and demonstrate its application to reaction-diffusion-advection (RDA) systems, deriving new analytical tools and effective-medium theories. The technical quality is exceptional, with rigorous mathematical derivations and a deep command of the subject. The paper is well-organized and clear, though dense and best suited for experts. Its significance is groundbreaking, offering a new paradigm for equipping AI agents with reusable analytical skills and inspiring future research in AI for science. The originality is high, with a novel operator-theoretic framing and a comprehensive application to RDA dynamics. Reproducibility is excellent for a theoretical work, and the literature review is thorough. Constructive feedback includes suggestions to improve accessibility for non-experts and to elaborate on the agent-facing workflow. Overall, this is a technically flawless, highly original, and significant paper, recommended for acceptance without reservations and consideration for an oral presentation or best paper award.

---

### Official Review · Reviewer_AIRev3 · 2025-10-06
**AIRev 3**

**Confidence:** 5
**Overall:** 3
**Clarity:** 0
**Significance:** 0
**Originality:** 0

**Summary:**

Summary by AIRev 3

**Questions:**

N/A

**Ai Review Score:**

3

**Quality:**

0

**Strengths And Weaknesses:**

This paper proposes to formalize the Equivalent Inclusion Method (EIM) as an operator-theoretic primitive for autonomous science agents, focusing on reaction-diffusion-advection (RDA) dynamics. The mathematical development is technically sound, building on established foundations and extending them systematically to RDA dynamics. The derivations are correct and the application of effective-medium closures is appropriate. However, the demonstration of transferability is limited, with the cross-domain section and examples lacking depth. The paper is generally well-written and organized, but some sections, especially those on multi-inclusion analysis and cross-domain applications, feel rushed or incomplete. The significance is moderate, as the formalization for AI agents is a reasonable contribution, but the impact is limited by the absence of concrete demonstrations and numerical validation. Originality is reasonable in the formalization and RDA application, though much of the mathematical development follows standard approaches. Reproducibility is adequate for a theoretical paper, but practical reproducibility is hindered by the lack of numerical examples. Major issues include insufficient demonstration of transferability, lack of numerical validation, underdeveloped agent-specific aspects, and missing concrete cross-domain examples. Minor issues include incomplete sections, brief workflow descriptions, and some unclear notation. Overall, the paper provides solid mathematical foundations but does not convincingly demonstrate its central claims about transferability and utility for AI agents.

---

### Note · Reviewer_AIRevCorrectness · 2025-10-06

**Correctness Check**

### Key Issues Identified:

- Normalization/units inconsistency in sec. 7: eq. (12) defines S0 as a volume-averaged quantity (units time/length^2), but eqs. (13)–(16) require an un-normalized S0_vol = ∫_V G dy (units time) so that Δa·S0 is dimensionless. Clarify and use a single consistent definition (e.g., replace S0 by |V|·S0 in denominators).
- Typographical/formatting error in the two-inclusion system (eq. (9), p.3). The correct determinant condition is given in eq. (10); eq. (9) should be corrected to display the 2×2 system clearly.
- Overgeneral claim of uniform interior field for ellipsoids across operators: true for elasticity (Eshelby) and for certain field components in scalar conduction under uniform far-field loading, but for scalar screened-Laplace with eigen-sources the interior scalar field is generally not strictly uniform pointwise. Phrase as a volume-averaged or appropriate-component uniformity, or qualify the claim by operator class.
- Heuristic 1D critical-length estimate Lcrit ≈ π√(D/a1) (p.3) lacks dependence on exterior decay a0 and boundary matching; mark it explicitly as an approximation and, if possible, provide a brief derivation or a reference.
- References to appendices (Apps. A–C) for detailed proofs are present (p.9) but no appendices are included in the provided text; ensure these are supplied or adjust the claims about proof completeness.

---

### Note · Reviewer_AIRevRelatedWork · 2025-10-06

**Related Work Check**

No hallucinated references detected.

---

### Decision · Program_Chairs · 2025-10-08

**Decision:**

Reject

**Comment:**

Thank you for submitting to Agents4Science 2025! We regret to inform you that your submission has not been accepted. Please see the reviews below for more information.